# Physical Activity Programming Advertised on Websites of U.S. Islamic Centers: A Content Analysis

**DOI:** 10.3390/ijerph15112581

**Published:** 2018-11-18

**Authors:** David Kahan

**Affiliations:** School of Exercise and Nutritional Sciences, San Diego State University, 5500 Campanile Drive, San Diego, CA 92182-7251, USA; dkahan@sdsu.edu; Tel.: +1-619-594-3887; Fax: +1-619-594-6553

**Keywords:** physical activity, health promotion, Internet, religious institutions, Islam

## Abstract

Previous research has found churches to be effective at delivering physical activity (PA) programs to their congregants. Mosques, however, have not been extensively studied. Therefore, we quantified U.S. Islamic centers’ advertisement of PA programming and examined their programming characteristics. We conducted a content analysis of the websites of 773 eligible Islamic centers of which 206 centers in 32 states advertised PA programming. We categorized PA by program type: camping, fitness classes, sports, youth programs, and irregular offerings. We calculated descriptive statistics by program type for specific activity, frequency/duration/volume, participant/instructor sex, and instructor religion. Youth group (44%) and sports (23%) programs were most and least frequently advertised, respectively. Most centers (66%) that posted information on PA programming advertised only one program type. Men and Muslims taught most activities. Most activities—except for fitness classes—were advertised to a male audience. Islamic centers should offer and advertise additional PA programming—especially for women—and better utilize their websites for promoting such programming. Individual Islamic centers and Islamic- and non-religion based public health agencies can utilize our findings to fashion future PA offerings.

## 1. Introduction

The World Health Organization defines an adequate level of physical activity (PA) as 150 min/wk of moderate PA, 75 min of vigorous PA, or an equivalent combination [1]. There are many benefits from regular participation in PA, including risk reduction of some cancers, cardiovascular disease, diabetes, and metabolic syndrome [2]. Yet 23% of adults and 80% of school-going adolescents are considered to be physically inactive [3]. Physical inactivity disparities exist on a global level, including those based on wealth, urbanicity, sex, and age [1,4]. Physical inactivity disparity based on religion also exists: 38 Muslim countries were 1.2 times more likely to be physically inactive than 94 non-Muslim countries, and within the Muslim countries, women’s physical inactivity prevalence (35.5%) was 1.4 times greater than men’s [5]. Insufficient PA is a risk factor for overweight, diabetes, and cardiovascular disease, and (perhaps) consequently their prevalence is alarmingly and disproportionately high among various ethnic groups of Muslims and Muslim women living in the United States (U.S.) [6,7,8,9].

In the U.S., Muslims currently number approximately 3.45 million persons, and in 2012, permanent resident status was granted to 100,000 Muslim immigrants [10]. A number of studies conducted in the U.S. collectively suggest that Muslims, particularly women, generally face more barriers to participation in PA compared to non-Muslims and to conditions encountered in their Islamic birth/heritage country including issues related to access, scheduling, clothing, gender roles, normative beliefs, self-efficacy, knowledge, social support, neighborhood conditions, and others [11,12,13,14,15,16,17,18,19,20,21]. Cultural factors, including level of community and familial social support for Muslim women engaging in PA, may dictate the environmental/structural conditions necessary for their unimpeded access to PA (e.g., need for modest dress, discrete changing spaces, and single-sex PA classes, instructors, and supervision) [19,20,22,23].

The National Physical Activity Plan (NPAP) addresses many barriers to PA, and a key aspect of the NPAP is its focus on strategies, tactics, and objectives across nine sectors—one of which is faith-based settings [24]. Faith-based settings are thought to possess “unique social systems, environments and physical structures (e.g., fellowship halls), communication channels, policies and practices, and often, health-related goals and supports (i.e., health ministries), which make them particularly conducive to promoting physical activity” [24].

Among U.S. Muslims, 55% attend religious services at least once monthly [25]. It is thus reasonable to infer that an even greater percentage attend mosques for any reason over the same time span. Thus, multiple persons are theoretically available to support a congregant’s PA directly (e.g., co-participation) and indirectly (e.g., policy, programs) including co-religionists, clergy, and the congregation itself. A mosque’s clergy, known as imams, occupy a key position for promoting community health through role modeling, involvement in communal decision-making, and gatekeeping on relevant social and cultural issues [26]. In the U.S. specifically, Muslims may view “imams, and thereby mosques, [as] a venue through which community health may be enhanced, trust established, and healthcare disparities reduced” ([27], p. 368). As well, the Friday sermon, known as the khutbah, can be an effective medium for delivering health promotion messages [28]. Furthermore, a mosque’s physical environment and adjoining grounds may offer venues and equipment for performing PA.

While the argument for and appeal of mosques for promoting health seems tenable, there is little evidence to support this idea in western societies. Smoking cessation educational programs delivered by religious teachers at seven mosques in the United Kingdom were evaluated [29], but for PA behavior, we found only one program evaluation of a mosque-based exercise program [30]. Specifically, scores on self-efficacy, importance of engaging in regular PA, participation in PA, peak aerobic capacity, and functional quality of life improved among 62 Muslim women who participated in a 1-h, 3-times/wk, 6-month-long class that offered walking, resistance training, relaxation, and chair exercises held in the Sisters prayer area of one mosque [30].

Because intervention evidence is scant, we believe it imperative to collect additional data in order to better profile to what degree and how Muslim religious institutions in the U.S. advertise their PA programming. Such an endeavor is important given the: (1) ascribed potential for faith-based organizations in general to influence health behavior, (2) large role Islam plays in the lives of a majority of U.S. Muslims, (3) growing population of Muslims in the U.S., (4) plausibility of mosques to provide PA programming, and (5) barriers to PA that many Muslims, particularly women, face. Our research also aligns with NPAP Faith-Based Settings Strategy 1 Tactic 4: “Create an environment supportive of physical activity by delivering evidence-based physical activity messaging and programs that are consistent with the faith community’s religious beliefs” [24].

Contemplating a survey of Muslim religious institutions’ advertisement of PA programming made us aware of challenges facing religious and cultural outsiders working with this population. U.S. Muslims can be mistrustful of researchers outside the community and be reluctant to participate in research, and finding the contact person in the mosque with direct knowledge of the research topic under study can be difficult even for insiders [31]. Moreover, we wanted to obtain a complete national picture of the PA programming advertised, which necessitates high response rates. For these reasons, we chose the Internet as the preferred medium for data collection. There is consensus among American scholars of Islam that Internet use is permissible [32], and U.S. Muslim women use Internet media in the acculturative process [33]. Relying on web pages as the data source eliminates recall bias and social desirability of human respondents. (Previously, Internet webpages were content analyzed to examine the health promotion efforts of an entire country’s local health authorities [34].)

Our study’s primary purpose was to analyze U.S. Islamic centers’ web page content to ascertain the number and types of programs and PA advertised to their communities. Secondarily, we were interested in program availability (by sex), context, and dosage (minutes of PA offered), and demographics of program leaders for the various types of programs advertised.

## 2. Method

### 2.1. Sample Derivation

The *American Mosque 2011 Report* identified 2106 mosques with 2.6 million participants [35]. The patchwork methods used for data collection, however, suggest that there is no central repository for mosque locations in the U.S. Therefore, we consulted a database of masjids, mosques, and Islamic centers in the U.S. [36]. A message on the database webpage indicates that the list may not be exhaustive but is continually updated as new information is received. We focused on identifying Islamic centers or similarly named organizations over organizations referred to only as mosques/masjids, as the former implies an organization that offers a constellation of health, human, and social services while the latter implies an organization whose primary services are related to prayer [37]. The San Diego State University Institutional Review Board exempted the study from review because it collected publicly available data and required no human participants.

In the database, we examined all 50 states’ and the District of Columbia’s list of organizations state-by-state. At this juncture we identified 1419 organizations (Figure 1), which we reduced by 95 for not qualifying as an Islamic center (i.e., the words *association*, *center*, *society*, *community*, or *circle* were not in the titles of the organizations). We screened remaining centers and excluded 541 primarily for not having a website (*n* = 444), which resulted in an eligible sample of 773 centers (Figure 1). We then opened each center’s website and searched all accessible pages for advertisement of PA programming. The vast majority (73.6%) of eligible centers did not post information about PA programming. The final analytic sample then was composed of 206 Islamic centers representing 32 states with most centers located in California (*n* = 30), Texas (*n* = 21), New Jersey and New York (*n* = 17 each), and Illinois (*n* = 14).

### 2.2. Data Extraction

We extracted data over a 1-mo period (July–August, 2017), which coincided with the observance of Ramadan and its culminating Eid al Fitr. From the publicly accessible pages of each website, we copied all verbiage pertaining generally and specifically to PA programming including offerings, intended audience(s), dates/days and times, locations, and instructor information We then pasted verbiage into an Excel worksheet.

### 2.3. Data Classification

We read and sorted extracted data into five types of PA based on the type of program offered and its intended audience:(1)Camps included summer, weekend, and specialty camps open to families, adults, or youth.(2)Fitness classes included regularly meeting martial arts, exercise or fitness, or non-sport, classes open to families, adults, or youth.(3)Sports programs included regularly meeting practices, games, and leagues for individual, dual, and team sports open to adults or youth.(4)Youth group programs included those found on youth group pages or distinguished as intended for youth.(5)Irregular programs included those that were offered once (i.e., special event), infrequently/irregularly, or intermittently and open to families, adults, or youth.

### 2.4. Variable Derivation

For all five types of PA programs offered, offerings per center, offerings by activity name, and target participants per activity (men only, women only, other [coed/not reported/unclear]) were identified or calculated. For fitness classes and sports programs, activity duration (min), frequency (day/wk), volume (min/wk), and instructor sex and religion (Muslim or non-Muslim) were identified or calculated. (We identified an instructor as Muslim if referred to in verbiage as brother or sister or if having a Muslim forename [38] or surname [39].) Camp length (days) and irregular program context (i.e., conditions under which the program was offered), when reported, were also identified or calculated.

### 2.5. Statistical Analyses

We descriptively analyzed the data using Microsoft Excel, which included calculation of measures of central tendency and variability. We report frequencies, minimum and maximum values, median, and interquartile range because all variables were non-normally distributed based on interpretation of the K-S statistic.

## 3. Results

### 3.1. Overview

Results are presented in Table 1, Table 2, Table 3, Table 4 and Table 5. In each table, the frequency of centers reporting a particular variable datum is presented and may be less than the frequency of centers that advertised a program from a particular category: camp (*n* = 51, 25% of analytic sample of 206), fitness class (*n* = 53, 26%), sports program (*n* = 47, 23%), youth group activity (*n* = 90, 44%), and irregular program (*n* = 73, 35%). Most centers advertised programming from only one category (*n* = 135, 66%) followed by two (*n* = 47, 23%), three (*n* = 14, 7%), four (*n* = 7, 3%), and all five (*n* = 3, 1%) categories.

In each table, the column following the frequency of centers with data, reports cumulative offerings (i.e., some centers advertised more than one program or one program separated by participant age/sex in a particular category), with values that exceeded center frequency values. For each category of program, we present only specific activity offerings whose frequency was ≥ 5. (See Appendix A for a complete list.)

### 3.2. Camps

Fifty-one centers advertised 56 camps that included 162 physical activities (Table 1). Of the 10 activities with frequency ≥ 5, the most common PA was sports (generic term). Of the remainder, four were team sports and five were individual activities. There were more men-only than women-only camps (7 vs. 5). Few centers reported length, which ranged between 2 and 54 days (*Med* = 8.0 ± 16.0).

### 3.3. Fitness Classes

Fifty-three centers advertised 73 classes (Table 2). Three of the five activities with frequency ≥ 5 were specific martial arts (e.g., karate, taekwondo) and martial arts (generic term) classes. The remaining two activities were fitness (generic term) and pilates/yoga classes. There were nearly twice as many women-only classes than men-only classes (33 vs. 17). Fewer centers reported class duration or frequency. Class duration ranged between 30 min and 491 min (i.e., mean length of open swimming/day) (*Med* = 60.0 ± 60.0), and class frequency ranged between 0.25 day/wk (i.e., once monthly) and 7 day/wk (*Med* = 1.0 ± 1.0). Class volume was typically not reported: Our calculated values ranged between 30 and 3435 min/wk (*Med* = 120.0 ± 120.0). Muslim and male class instructors outnumbered non-Muslim and female class instructors by ratios of 10:1 and nearly 2:1, respectively.

### 3.4. Sports Programs

Forty-seven centers advertised 100 programs (Table 3). Four of the six activities with frequency ≥ 5 were team sports while the remainder were individual or dual sports. (The two most frequent activities—soccer and basketball—comprised 50% of all 100 programs reported.) There were nearly twice as many men-only than women-only programs (53 vs. 29). Few centers reported practice/game duration or session frequency. Practice/game duration ranged between 30 and 240 min/session (*Med* = 120.0 ± 60.0), and session frequency ranged between 0.25 day/wk (i.e., once monthly) and 4 day/wk (*Med* = 1.0 ± 1.0). Session volume ranged between 40 and 720 min/wk (*Med* = 180.0 ± 120.0). Muslim and male coaches outnumbered non-Muslim and female coaches by ratios of 8:1.

### 3.5. Youth Group Activities

Ninety centers advertised 141 physical activities (Table 4). Of the seven activities with frequency ≥ 5, the most common was sports (generic term). Of the remainder, two were team sports and four were individual, dual, or group activities. There were 1.4 times more men-only than women-only offerings (36 vs. 26).

### 3.6. Irregular Programs

Seventy-three centers advertised 122 physical activities (Table 5). Of the 5 activities with frequency ≥ 5, 3 were team sports, 1 was sports (generic term), and 1 was run/walk. When a context for an activity was provided, most offerings (68%) were specifically described (i.e., time, place, etc.), with Eid/Ramadan being the most common context with frequency ≥ 5. There were 1.4 times more men-only than women-only offerings (36 vs. 26).

## 4. Discussion

U.S. religious institutions occupy a key role in promoting and providing PA to their congregants [24]. Mosques, in particular, may draw women who are reluctant to exercise in mixed-sex settings because they: (1) are already a part of such persons’ lifestyle, (2) may readily configure PA programming to abide by cultural/religious norms, and (3) offer socializing—through such programs—that strengthens social bonds and reinforces righteous behavior [40]. Our findings—based on analysis of 773 US Islamic centers’ webpage content—suggest that a large majority (73%) elected not to advertise PA programming on their websites. We are unsure why so many did not advertise PA programming on their websites but possibilities include (1) not offering any PA programs, (2) subordinating PA relative to other content (e.g., prayer times, *salat* (how to pray), FAQs about Islam), and (3) posting events with PA on members-only media platforms such as Facebook.

For the 206 Islamic centers that advertised PA programming, we found a large variety of PA options of varying frequency/duration. PA in camp settings included many moderate-to-vigorous PA (MVPA) options such as sports, soccer, swimming, and hiking. This finding was encouraging because camps are acknowledged for increasing summertime PA and decreasing sedentary behavior among youth [41]. PA in fitness class settings included many martial arts forms, which have become more popular, especially among females, within the Muslim community in recent years in response to anti-Islamic sentiment [42]. Along with fitness and pilates/yoga classes, however, the median cumulative PA volume of 120 min/wk fell short of the recommendation of 150 min/wk of MVPA [1]. The fitness class setting was the only one for which postings of single-sex classes for women outnumbered those for men (i.e., 2 to 1), yet twice as many men vs. women taught fitness classes. It is important to consider the sex of class relative to overcoming barriers to participation—particularly for women [22]. Over 90% of listed instructors were Muslim, which may attract participants as well as offer co-religionist modelling of and motivation for a physically active lifestyle.

PA in sports settings primarily consisted of basketball and soccer but others, such as badminton and cricket, are popular in South Asian countries, which represent the origins of 35% of foreign-born U.S. Muslims [25]. As stated, we found similar issues related to coach sex and coach religion. Unlike fitness classes, median cumulative PA volume for sport practices and games of 180 min/wk exceeded the recommendation of 150 min/wk of MVPA [1]. Websites advertised 1.8 times more sport opportunities for men than women. The growth of and interest in Muslim women’s sport worldwide [43] suggests that Islamic centers should promote additional sport opportunities for women as well as make accommodations to their physical environment and spaces in order to foster more equitable access.

PA in youth group settings included several sports found in other categories, outdoor activities such as hiking and canoeing/kayaking, as well as bowling. Websites advertised 1.4 times more youth program PA opportunities for boys than girls and similar recommendations proposed for sport programs should be considered. Out of 18 aspects of the mosque, U.S. imams assigned the lowest performance grade to involvement in youth activities and 61% listed increasing youth activities among their top three priorities [44]. Overall, youth vs. adult participation in mosque activities is lower, and U.S. Islamic centers have been encouraged to expand program offerings in order to bridge a cultural and generational divide for attracting and retaining youth engagement [45]. From a health perspective, Islamic centers are enjoined to provide PA opportunities to their youth [46] and some mosques have provided youth PA/sport opportunities to attract youth and create socialization opportunities for them [19].

For irregular programs, sport activities and run/walk activities were typically advertised in the context of tournaments and greater community health and wellness events, respectively. Websites advertised 1.4 times more irregular programs for men than women and similar recommendations proposed for sport and youth programs should be considered. Irregular programs were most frequently associated with Eid and Ramadan observances, which draw a larger segment of the Muslim community to an Islamic center.

National organizations such the U.S. Council of Muslim Organizations, American Muslim Health Professionals, and Islamic Medical Association of North America, as well as relevant regional and community organizations should consider developing targeted messaging, and workshops and trainings that assist Islamic centers in PA program development and promotion in general, which in turn could be advertised on websites. Doing so is particularly important in light of our findings and the prevalence of morbidity associated with insufficient PA found among Muslims in the U.S. [6,7,8,9]. As well, imams, who play a major role in matters related to Muslim community health, should also receive training for how to incorporate PA promotion into day-to-day ministry and chaplaincy activities [26,27,47] and ensure that PA programming is sufficiently advertised. Most U.S. mosques are not considered women-friendly; thus, it is important moving forward that women are purposely included in decisions related to governance and development of women’s and family PA programs [48]. To the degree required by a local Islamic community in general, and particularly its women, individual centers should strive to offer conditions that facilitate women’s unfettered involvement in PA programs such as providing women-only classes, female instructors, and discrete spaces for PA [22,23].

Strengths of the study include its national scope and its analysis of websites, which allows for public scrutiny of PA content that may not be accessible to community outsiders vis-à-vis standard questionnaire/interview protocol. Our analysis was limited to the PA programs advertised on websites, which may have resulted in missing programs that were not specifically advertised on websites. Capturing non-advertised PA programs would entail access to social media platforms and Islamic center personnel, which could have posed logistical challenges. Meanwhile, for informational and health promotion purposes, we encourage more Islamic centers to post PA programs on their websites.

Our study is the first to examine the content and characteristics of PA programming offered by U.S. Islamic centers as advertised on their websites. Future studies can use our data as a baseline to determine changes over time with respect to the variables studied. We recommend that Islamic center personnel and community health organizations working with this data: (1) comprehensively audit center websites for inclusion of PA programming, and (2) systematically devise, offer, and promote additional PA programming.

### Specific Recommendations

Multiple mechanisms for achieving recommendation (2) in the previous paragraph are available. For example, Islamic centers could form a health and wellness council comprised of youth, men, and women charged with advising on, developing, and implementing PA programming. Additionally, it is important to build PA programming into the natural cycle of events in the Islamic calendar and at Islamic centers, which could include PA programming before/after the weekly Friday Jummah prayer, youth Sunday school, and Ramadan Iftar meals. The month of Rabi’ al-awwal (Prophet Muhammad’s birth month) could be designated for multiple month-long PA programs in accordance with hadith (collection of the sayings attributed to the Prophet) that ascribe his promotion of swimming, archery, and horseback riding.

When an Islamic center is limited in physical space, funds, or expertise to organize/lead PA programs multiple resolutions may be available. For example, in locales with large Muslim populations, putting on joint programming with other Islamic centers may be feasible. Also, universities may offer community-based participatory research opportunities to the Muslim community that target PA [49]. Nonprofit community organizations such as the YMCA may also offer culturally tailored programming—such as women-only swims for Muslims [50]. Lastly, community-based cultural organizations whose membership may include Muslims may offer PA programming solely or in partnership with other community organizations. For example, in San Diego, the Dunya Women’s Health Collaborative offers women-only yoga and swimming classes and a girls-only basketball league at the neighborhood YMCA [51]. For these and other examples, it is imperative for imams to promote and publicize them to congregants, on websites, and on social media.

## Figures and Tables

**Figure 1 ijerph-15-02581-f001:**
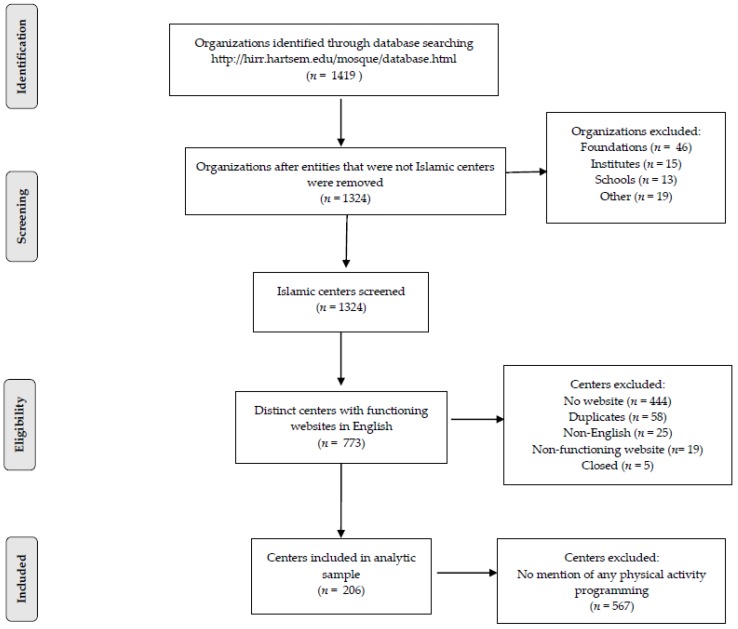
Flow diagram depicting steps from population of Islamic organizations to analytic sample of Islamic centers.

**Table 1 ijerph-15-02581-t001:** Islamic center camp descriptive data (*n* = 51 centers).

Category	Centers with Data (*n*)	Cumulative Camps (*n*)	Freq (%)	*Min*	*Max*	*Med*	*IQR*
Activities	51	56 ^a^		1.0	12.0	2.0	3.0
Sports ^b^			23 (14.2)				
Soccer			16 (9.9)				
Swimming			15 (9.2)				
Hiking			14 (8.6)				
Basketball			13 (8.0)				
Canoeing/Kayaking			12 (7.4)				
Archery			9 (5.6)				
Climbing			5 (3.1)				
Cricket			5 (3.1)				
Volleyball			5 (3.1)				
Other activities ^c^			45 (27.8)				
Camp length (day)	41	47		2.0	54.0	8.0	16.0
Camp participants	51	58					
Men only			7 (12.1)				
Women only			5 (8.6)				
Other ^d^			46 (79.3)				

^a^ Cumulative camps listed 162 activities. ^b^ Sports used as generic descriptor without identifying specific activity. ^c^ Represents 29 unique activities for which cumulative frequency < 5 each. ^d^ Mixed sex/sex not reported or unclear.

**Table 2 ijerph-15-02581-t002:** Islamic center fitness class descriptive data (*n* = 53 centers).

Category	Centers with Data (*n*)	Cumulative Classes (*n*)	Freq (%)	*Min*	*Max*	*Med*	*IQR*
Activities	53	73		1.0	5.0	1.0	0.0
Karate			13 (17.6)				
Taekwondo			10 (13.5)				
Fitness ^a^			9 (12.2)				
Pilates/Yoga			7 (9.5)				
Martial arts ^b^			5 (6.8)				
Other activities ^c^			30 (40.5)				
Class duration (min/class)	35	54		30.0	491.0	60.0	60.0
Class frequency (class/wk)	43	61		0.25	7.0	1.0	1.0
Volume (min/wk)	53	57		30.0	3435.0	120.0	120.0
Class instructor sex	27	41					
Male			27 (65.9)				
Female			14 (34.1)				
Class instructor religion	24	33					
Muslim			30 (90.9)				
Non-Muslim			3 (9.1)				
Class participants	53	88					
Women only			33 (37.5)				
Men only			17 (19.3)				
Other ^d^			38 (43.2)				

^a^ Fitness includes the phrasings fitness center, fitness class, group fitness, open gym. ^b^ Website used Martial arts as generic descriptor without identifying specific type. ^c^ Represents 13 unique activities for which cumulative frequency < 5 each. ^d^ Mixed sex/sex not reported or unclear.

**Table 3 ijerph-15-02581-t003:** Islamic center sports program descriptive data (*n* = 47 centers).

Category	Centers with Data (*n*)	Cumulative Programs (*n*)	Freq (%)	*Min*	*Max*	*Med*	*IQR*
Programs	47	100		1.0	8.0	1.0	2.0
Soccer			27 (27.0)				
Basketball			23 (23.0)				
Badminton			11 (11.0)				
Volleyball			8 (8.0)				
Cricket			5 (5.0)				
Table tennis			5 (5.0)				
Other activities ^a^			21 (21.0)				
Practice/game duration (min/session)	26	61		30.0	240.0	120.0	60.0
Practice/game frequency (session/wk)	33	79		0.25	4.0	1.0	1.0
Volume (min/wk)	24	57		40.0	720.0	180.0	120.0
Sport coach sex	7	9					
Male			8 (88.9)				
Female			1 (11.1)				
Sport coach religion	7	9					
Muslim			8 (88.9)				
Non-Muslim			1 (11.1)				
Program participants	47	151					
Men only			53 (35.1)				
Women only			29 (19.2)				
Other ^b^			69 (45.7)				

^a^ Represents 11 unique sports for which cumulative frequency < 5 each. ^b^ Mixed sex/sex not reported or unclear.

**Table 4 ijerph-15-02581-t004:** Islamic center youth group activity descriptive data (*n* = 90 centers).

Category	Centers with Data (*n*)	Cumulative Activities (*n*)	Freq (%)	*Min*	*Max*	*Med*	*IQR*
Programs	90	141		1.0	8.0	1.0	0.0
Sports/athletics ^a^			25 (17.7)				
Hiking			24 (17.0)				
Basketball			19 (13.5)				
Bowling			12 (8.5)				
Laser tag/mini-golf/paintball			12 (8.5)				
Soccer			7 (5.0)				
Canoeing/kayaking			5 (3.5)				
Other activities ^a^			37 (26.2)				
Program participants	90	147					
Men only			36 (24.5)				
Women only			26 (17.7)				
Other ^c^			85 (47.8)				

^a^ Sports or athletics used as generic descriptor without identifying specific activity. ^b^ Represents 21 unique activities for which cumulative frequency < 5 each. ^c^ Mixed sex/sex not reported or unclear.

**Table 5 ijerph-15-02581-t005:** Islamic center irregular program descriptive data (*n* = 73 centers).

Category	Centers with Data (*n*)	Cumulative Programs (*n*)	Freq (%)	*Min*	*Max*	*Med*	*IQR*
Programs	73	122		1.0	8.0	1.0	1.0
Soccer			19 (15.6)				
Basketball			18 (14.8)				
Sports/athletics ^a^			17 (13.9)				
Run/walk			10 (8.2)				
Football			6 (4.9)				
Other activities ^b^			52 (42.6)				
Context (offerings/center)	73	84					
General			27 (32.1)				
Specific			57 (67.9)				
Eid/Ramadan			17 (29.8)				
Tournament			15 (26.3)				
Community run/race walk			11 (19.3)				
Other contexts ^c^							
Program participants	73	147					
Men only			36 (24.5)				
Women only			26 (17.7)				
Other ^d^			85 (47.8)				

^a^ Sports or athletics used as generic descriptor without identifying specific activity. ^b^ Represents 26 unique activities for which cumulative frequency < 5 each. ^c^ Represents 10 unique contexts for which cumulative frequency < 5 each. ^d^ Mixed sex/sex not reported or unclear.

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
