# Peer review of "Physical Activity Programming Advertised on Websites of U.S. Islamic Centers: A Content Analysis"

_ijerph, 2018, doi:10.3390/ijerph15112581_

Round 1
Reviewer 1 Report
A good paper. The weakness of the paper is in its conclusions which with a little work could make a very good paper.
Clear recommendations/direction should be given on how to use this paper in future programs. Language such as ' Our findings may stimulate Islamic center personnel' and 'public health agencies can utilize our findings to fashion future PA offerings' is too passive.
Please separate out the key recommendations at the end of the paper.
Author Response
Thank you for your overall positive review of our manuscript and for your helpful suggestions.
A number of specific directions and recommendations have been added to the revised manuscript, which are highlighted in red font (lines 310-331 of revision).
The exact verbiage appears below:
4.1. Specific Recommendations
Multiple mechanisms for achieving the latter recommendation are available. For example, Islamic centers could form a health and wellness council comprised of youth, men, and women charged with advising on, developing, and implementing PA programming. Additionally, it is important to build PA programming into the natural cycle of events in the Islamic calendar and at Islamic centers, which could include PA programming before/after the weekly Friday Jummah prayer, youth Sunday school, and Ramadan Iftar meals. The month of Rabi’ al-awwal (Prophet Muhammad’s birth month) could be designated for multiple month-long PA programs in accordance with hadith (collection of the sayings attributed to the Prophet) that ascribe his promotion of swimming, archery, and horseback riding.
When an Islamic center is limited in physical space, funds, or expertise to organize/lead PA programs multiple resolutions may be available. For example, in locales with large Muslim populations, putting on joint programming with other Islamic centers may be feasible. As well, universities may offer community-based participatory research opportunities to the Muslim community that target PA [49]. Nonprofit community organizations such as the YMCA may also offer culturally tailored programming – such as women-only swims for Muslims [50]. Lastly, community-based cultural organizations whose membership may include Muslims may offer PA programming solely or in partnership with other community organizations. For example, in San Diego the Dunya Women’s Health Collaborative offers women-only yoga and swimming classes and a girls-only basketball league at the neighborhood YMCA [51]. For these and other examples, it is imperative for imams to promote and publicize them to congregants, on websites, and on social media.
Associated new references:
49. UC San Diego School of Medicine Center for Community Health. Faith-Based Wellness: Addressing Health Disparities in African American, Latino and Muslim Communities in San Diego Website. Available from: https://ucsdcommunityhealth.org/work/faith-based-wellness/ (Accessed on 13 November 2018).
50. YMCA of San Diego County. Y Women-Only Swim for Muslim Women Website. Available from: https://www.ymca.org/about-y/news-center/programs/y-women-only-swim-muslim-women. Published on 27 June 2012 (Accessed on 13 November 2018).
51. United Women of East Africa. Dunya Women’s Collaborative Website. Available from: https://www.uweast.org/category/health-wellbeing/ (Accessed on 13 November 2018).
Reviewer 2 Report
I think this is an excellent paper and provides a very informative context of the state of physical activity programming in faith based settings for the Muslim community.
My only feedback is to include a couple of additional points in the background literature:
1) There is extensive detail regarding the physical inactivity of Muslim men and women however, no discussion of the health risks associated with this physical inactivity. This is especially relevant due to the higher levels of diabetes and heart disease that are seen in the South Asian population where a large proportion of the Muslim community have ethnic roots.
2) I believe there needs to be more detail about the challenges that women face with engaging in physical activity in the Muslim community. Specific references to the need for sex based classes due to religious background should be addressed and then linked into the discussion where you found lower numbers of female fitness instructors than males.
Author Response
Thank you for your overall positive review of our manuscript and for your suggestions.
Your comments:
My only feedback is to include a couple of additional points in the background literature:
1) There is extensive detail regarding the physical inactivity of Muslim men and women however, no discussion of the health risks associated with this physical inactivity. This is especially relevant due to the higher levels of diabetes and heart disease that are seen in the South Asian population where a large proportion of the Muslim community have ethnic roots.
2) I believe there needs to be more detail about the challenges that women face with engaging in physical activity in the Muslim community. Specific references to the need for sex based classes due to religious background should be addressed and then linked into the discussion where you found lower numbers of female fitness instructors than males.
Our response:
1) Verbiage related to health risks due to insufficient PA that affect Muslim Americans has been added. See lines 35-38 and 285-286 (red font) in revised manuscript.
Specific verbiage includes:
Lines 35-38: Insufficient PA is a risk factor for overweight, diabetes, and cardiovascular disease, and (perhaps) consequently their prevalence is alarmingly and disproportionately high among various ethnic groups of Muslims and Muslim women living in the United States (U.S.) [6–9].
Lines 285-286: Doing so is particularly important in light of our findings and the prevalence of morbidity associated with insufficient PA found among Muslims in the U.S. [6–9].
Associated new references include:
6. Abuelezam, N.N.; El-Sayed, A.M.; Galea, S. The health of Arab Americans in the United States: an updated comprehensive literature review, Front. Public Health 2018, 6, 262.
7. Kanaya, A.M.; Herrington, D.; Vittinghoff, E.; Ewing, S.K.; Liu, K.; Blaha, M.J.; Dave, S.S.; Qureshi, F.; Kandula, N.R. Understanding the high prevalence of diabetes in U.S. South Asians compared with four racial/ethnic groups: the MASALA and MESA studies, Diabetes Care 2014, 37, 1621–1628.
8. Nieru, J.W.; Tan, E.M.; St. Sauver, J.; Jacobson, D.J.; Agunwamba, A.A.; Wilson, P.M.; Rutten, L.J.; Damodaran, S.; Wieland, M.J. High rates of diabetes mellitus, pre-diabetes and obesity among Somali immigrants and refugees in Minnesota: a retrospective chart review, J. Immigr. Minor. Health 2016, 18, 1343–1349.
9. Budhwani, H.; Borgstede, S.; Palomares, A.L.; Johnson, R.B.; Hearld, K.R. Behavior and risks for cardiovascular disease among Muslim women in the United States, Health Equity 2018, 2, 264–281.
2) Several passages have been added in the background and discussion sections to address the unique needs of Muslim women related to PA have been added.
Specific verbiage includes:
Lines 45-48: Cultural factors, including level of community and familial social support for Muslim women engaging in PA, may dictate the environmental/structural conditions necessary for their unimpeded access to PA (e.g., need for modest dress, discrete changing spaces, and single-sex PA classes, instructors, and supervision) [19,20,22,23].
Lines 292-295: To the degree required by a local Islamic community in general, and particularly its women, individual centers should strive to offer conditions that facilitate women’s unfettered involvement in PA programs such as providing women-only classes, female instructors, and discrete spaces for PA [22,23].
Associated new reference includes:
23. UC San Diego’s Center for Community Health [UCSD-CCH]. Improving Muslim Youth Participation in Physical Education and Physical Activity in San Diego County; UCSD-CCH: San Diego, CA, USA, 2017. Available from: https://ucsdcommunityhealth.org/wp-content/uploads/2017/12/Final1_CLS_MuslimGuide9interactive.pdf (Accessed on 09 November 2018).